

# How training loads in the preparation and competitive period affect the biochemical indicators of training stress in youth soccer players?

Marcin Andrzejewski[1,*], Marek Konefał[2], Tomasz Podgórski[3], Beata Pluta[1], Paweł Chmura[4], Jan Chmura[2], Jakub Marynowicz[5], Kamil Melka[6], Marius Brazaitis[7] and Jakub Kryściak[3,*]

[1] Department of Methodology of Recreation, Poznań University of Physical Education, Poznań, Poland
[2] Department of Biological and Motor Sport Bases, Wroclaw University of Health and Sport Sciences, Wrocław, Poland
[3] Department of Physiology and Biochemistry, Poznań University of Physical Education, Poznań, Poland
[4] Department of Team Games, Wroclaw University of Health and Sport Sciences, Wrocław, Poland
[5] Department of Theory and Methodology of Team Sport Games, Poznań University of Physical Education, Poznań, Poland
[6] Institute of Mathematics, University of Wrocław, Wrocław, Poland
[7] Institute of Sports Science and Innovation, Lithuanian Sports University, Kaunas, Lithuania
* These authors contributed equally to this work.

Corresponding author
Jakub Kryściak,
j.krysciak@awf.poznan.pl

## ABSTRACT

**Background:** Physical fitness optimization and injury risk-reducing require extensive monitoring of training loads and athletes' fatigue status. This study aimed to investigate the effect of a 6-month training program on the training-related stress indicators (creatine kinase – CK; cortisol – COR; serotonin – SER; brain-derived neurotrophic factor – BDNF) in youth soccer players.

**Methods:** Eighteen players (17.8 ± 0.9 years old, body height 181.6 ± 6.9 cm, training experience 9.7 ± 1.7 years) were blood-tested four times: at the start of the preparation period (T0), immediately following the preparation period (T1), mid-competitive period (T2), and at the end of the competitive period (T3). CK activity as well as concentrations of serum COR, SER and BDNF were determined. Training loads were recorded using a session rating of perceived exertion (sRPE).

**Results:** Statistical analyzes revealed significant effects for all biochemical parameters in relation to their time measurements (T0, T1, T2, T3). The statistical analyzes of sRPE and differences of biochemical parameters in their subsequent measurements (T0–T1, T1–T2, T2–T3) also demonstrated significant effects observed for all variables: sRPE ($H^{KW}$ = 13.189 (df = 2); $p$ = 0.00), COR ($H^{KW}$ = 9.261 (df = 2); $p$ = 0.01), CK ($H^{KW}$ = 12.492 (df = 2); $p$ = 0.00), SER ($H^{KW}$ = 7.781 (df = 2); $p$ = 0.02) and BDNF ($H^{KW}$ = 15.160 (df = 2); $p$ < 0.001).

**Discussion:** In conclusion, it should be stated that the most demanding training loads applied in the preparation period (highest sRPE values) resulted in a significant increase in all analyzed biochemical training stress indicators. The reduction in the training loads during a competitive period and the addition of recovery training

sessions resulted in a systematic decrease in the values of the measured biochemical indicators. The results of the study showed that both subjective and objective markers, including training loads, are useful in monitoring training stress in youth soccer players.

# INTRODUCTION

During pre-season and competitive periods of the season, youth soccer players are exposed to heavy training and competitive workloads (*Nedelec et al., 2013*; *Becker et al., 2020*), which may be compounded by mental stress resulting from championship matches (*Meister et al., 2013*). Some authors reported that fatigue and decreased performance during soccer match are caused by disturbed muscle ion homeostasis, changes in internal muscle temperature, and decreased muscle glycogen level (*Mohr, Krustrup & Bangsbo, 2005*; *Bangsbo, Mohr & Krustrup, 2006*). The imbalance between fatigue and recovery may result in a decrease in exercise capacity and an increased risk of injuries related to overtraining syndrome (*Gabbett, 2016*; *Malone et al., 2015*). In order to correctly balance training and recovery, it is necessary to monitor individual fatigue levels and to adjust training loads accordingly (*Gabbett, 2016*; *Buchheit, 2014*; *Arazi et al., 2020*). The training loads of youth soccer players can be quantified by external parameters (*e.g.*, global position system (GPS) and video analysis) as well as internal parameters (*e.g.*, rate of perceived exertion (RPE), heart rate (HR), and blood lactate concentration) (*McLaren et al., 2018*; *Marynowicz et al., 2020*).

One of the most versatile and cost-effective methods for internal load monitoring, closely correlated with HR and blood lactate concentration, is the RPE (*Scott et al., 2013*). It integrates afferent neural signals from different inputs to the central nervous system and therefore provides a more comprehensive evaluation of internal training load at an individual level than, for example, HR measurements (*de Morree, Klein & Marcora, 2012*; *Abbiss et al., 2015*). *Foster et al. (2001)* proposed session rating of perceived exertion (sRPE) as a method for internal training load monitoring based on the category ratio Borg scale (CR-10) (*Borg, Hassmén & Lagerström, 1987*). Research indicates that sRPE might be a better suited and more essential method to monitor training youths in a sport like soccer (as opposed to monitoring training adults) (*Capranica & Millard-Stafford, 2011*). Some studies also highlighted relationships between sRPE and biochemical markers of muscle damage and chronic stress, such as creatine kinase activity and cortisol concentration (*Moreira et al., 2013*; *Caetano Júnior, Castilho & Raniero, 2017*; *Tiernan et al., 2020*).

Nowadays, to optimize physical fitness and to reduce the risk of injury, professional soccer clubs monitor their players' exercise stress level related to match and training loads by using a variety of biochemical, hematological, immunological, and endocrine parameters (*Becker et al., 2020*; *Becatti et al., 2017*; *Silva et al., 2018*; *Walker et al., 2019*;

*Andrzejewski et al., 2021*). The psycho-physiological process that causes fatigue is complex and determined by peripheral and central factors. Reduction in muscle pH, depletion of phosphocreatine and glycogen in muscle tissue, ammonia accumulation in the blood and tissues, muscle damage, and oxidative stress may cause a loss in the ability to generate muscle force (*Mohr, Krustrup & Bangsbo, 2005*; *Taylor, Todd & Gandevia, 2006*; *Parry-Billings et al., 1990*; *Sewell, Gleeson & Blannin, 1994*; *Davis, 2008*; *Finsterer, 2012*). Some of the most used and well-documented muscle damage marker is creatine kinase (*Stajer, Vranes & Ostojic, 2020*). Muscle damage very often initiates inflammation which can lead to prolonged, chronic stress and reduced exercise capacity (*Meister et al., 2013*; *Romagnoli et al., 2016*). Typically, endocrine marker of chronic stress is cortisol (*Tiernan et al., 2020*; *Andrzejewski et al., 2021*). This steroid hormone produced by the adrenal cortex, stimulates gluconeogenesis, glycogen synthesis, and protein synthesis in the liver. A high cortisol concentration may cause inhibition of the immune system and proteolysis, and for this reason it is related to the catabolic stress in the body (*Michailidis, 2014*). In practice, cortisol has been stated as reliable marker of training stress and is considered as an important hormones in the biochemical assessment of athletes (*Hackney, 2020*; *Saidi et al., 2020*).

Training stress is perceived also in the central nervous system (CNS) and is associated with disturbances in neurotransmitter synthesis including that of serotonin (*Cordeiro et al., 2014*), dopamine (*Watson et al., 2005*; *Foley & Fleshner, 2008*), acetylcholine (*Rodrigues et al., 2009*) and other neurotransmitters. *Newsholme, Acworth & Blomstrand (1987)* suggested that, during exercise, increased brain serotonergic activity may intensify lethargy and loss of drive, resulting in a reduction of motor unit recruitment. An increase in CNS concentrations of serotonin during exercise also leads to an increase in perceived exertion, likely by modifying the player's tolerance to pain or discomfort (*Meeusen, Watson & Dvorak, 2006*; *Meeusen & Roelands, 2018*).

However, suggesting that serotonin by itself reduces physical performance is too narrow as an explanation. *Davis & Bailey (1997)* found that an increase in serotonin activity during physical exercises inhibits dopamine, which is involved in movement initiation (*Chaudhuri & Behan, 2000*). Another external regulator of dopamine, in addition to serotonin, is brain-derived neurotrophic factor (BDNF). *Rojas Vega et al. (2006)* reported that serum BDNF transiently increased at the point of exhaustion during incremental exercise in healthy male athletes. In addition, long-term regular aerobic training was reported to increase the serum BDNF concentration at rest and even during 2 days of recovery (*Jeon & Ha, 2015*). Another important finding is that stress-induced hyperactivity of the hypothalamic-pituitary-adrenal (HPA) axis may decrease BDNF production *via* increased cortisol secretion and activation of the proinflammatory response (*Rothman & Mattson, 2013*). BDNF participates also in many physiological processes such as glycogen homeostasis and lipid metabolism (*Jiménez-Maldonado et al., 2018*). Many authors emphasize the importance of glycogen metabolism in soccer (*Mohr, Krustrup & Bangsbo, 2005*; *Bangsbo, Mohr & Krustrup, 2006*). Therefore, BDNF may play an important role in regulation of training and matches metabolism in soccer. Although the mechanisms
involved in the training-related stress development are now being recognized and explored, much of the picture remains unclear.

It is, however, well known that the applied training loads of athletes cause changes in the indicators that characterize the levels of training-related stress (*Silva et al., 2018*; *Cordeiro et al., 2017*). Yet, to our knowledge, no studies have assessed the relation between central and peripheral factors that characterize exercise-related stress and the assessment of perceived effort (sRPE) in youth soccer players. Obtaining a better understanding of the relation between these indicators (that characterize internal training loads) could help coaches increase training requirements and monitor the training process of athletes.

The aim of this study was therefore to investigate the effect of the applied training loads in a 6-month training program on the indicators characterizing the level of training-related stress in youth soccer players. In addition, the relation between internal biochemical parameters characterized by training loads and the assessments of perceived effort (sRPE) in the analyzed training cycle was investigated.

Therefore, it was hypothesized that there are positive relations between the applied training loads expressed as the sRPE value and biochemical markers characterizing training stress in youth soccer players.

## MATERIALS AND METHODS

### Experimental approach

The aim of this longitudinal study was to analyze the effects of the applied training loads on biochemical markers (creatine kinase, cortisol, serotonin and BDNF) characterizing the level of training-related stress throughout a 6-month soccer training period (January–June). Moreover, a potential relation between these indicators and sRPE during youth soccer training sessions was investigated.

### Sample collection

The players reported to a certified biochemical laboratory for the drawing of blood samples (see 'Blood collection and biochemical measurements' section below) during four time points throughout the season. Pre-season samples were drawn on the first day of practice, with players having refrained from training for 36 h (T0); subsequent blood collections were conducted every 7 weeks (approximately 36 h after a game) until the last competitive match (T3) (Fig. 1). The day after each match was reserved for passive recovery. Athletes, euhydrated and having fasted for at least 8 h, arrived between 07:00 and 09:00 a.m. (*Walker et al., 2019*).

In addition to biochemical analyzes, sRPE was recorded after each training session using a modified CR-10 Borg scale to subjectively assess the intensity of individual player's exercise during sessions and matches. In an attempt to assess the players' fatigue status at the start of the next training microcycle, the evaluation time points were planned as not to assess acute fatigue after a soccer match.

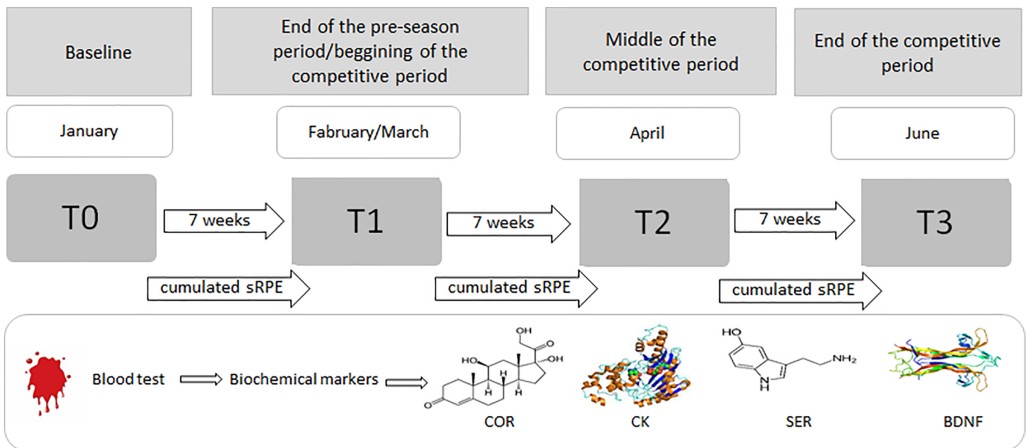

**Figure 1 Procedure for the blood sampling and collecting sRPE data during the 6-month period (January–June).** COR, serum cortisol concentration; CK, serum creatine kinase activity; SER, serum serotonin concentration; BDNF, serum brain-derived neurotrophic factor concentration.

## Subjects

Twenty-nine youth male soccer players from the highest division professional club in Poland participated in the study. Due to injuries sustained by some of the players during training and soccer matches, only the results of 18 players (17.8 ± 0.9 years old, body height 181.6 ± 6.9 cm), who completed the training program without sustaining serious injuries were analyzed. The inclusion criteria were that players participated in 90% of all training sessions and that they trained at the club for a full 6-month period and participated in each championship match for a minimum of 75 min per match.
The participants had significant soccer-training experience (9.7 ± 1.7 years) and were competing at the highest level in their under-19 age category. Their weekly training schedule involved five sessions on a soccer field (∼400 min/week) and one friendly or competitive match. Goalkeepers were excluded from the study since they did not take part in the same training sessions as the outfielders.

All players remained in the soccer center during the season. Under consented obligation, they all declared that they did not introduce any changes with regards to additional nutrition, medication, or supplementation. Consequently, their nutrition and hydration habits were unified, following the recommendations by the soccer center nutritional staff (*Mielgo Ayuso et al., 2017*). The players followed the same nutritional and hydration protocol during the soccer season. For this reason we belive that this aspects did not influence on registered indices. A detailed analysis of nutrition related to the supply of macronutrients and energy intake has not been performed. All participants were given a full explanation of the proposed study and its requirements. They were informed of potential risks and were provided with written consent forms. Additionally, for participants under 18 years of age, parental or guardian consent was required. Participants were free to withdraw at any time, without any repercussions. The study was conducted

according to a protocol (no. 973/17) that had been fully approved by the Poznan University of Medical Sciences ethical review board at the institution prior to the commencement of the study. The study furthermore conformed with the Declaration of Helsinki guidelines, and all health and safety procedures were complied with.

## Baseline measures

At the end of the autumn round, the players went on a 14-day passive rest break. During this period, they did not perform any individual training sessions. On the first day after returning from the break, the athletes were subjected to medical examinations. The next morning, as stipulated by the research schedule, the first tests were performed to assess the concentration of biochemical and endocrine markers in the players' blood. This marked the start of the pre-season period (for the spring round) and the first training session was initiated. Exercise intensity monitoring (sRPE) was implemented from the first training session by using the modified CR-10 Borg scale (T0–T1, T1–T2, and T2–T3). All data obtained from the blood samples (T1, T2, and T3) were compared to the data from the first week of the study (T0).

## Blood collection and biochemical measurements

Blood samples were collected by qualified, professional laboratory personnel. All safety precautions were strictly adhered to during the blood collection process. With the player comfortably seated, blood samples were collected from the fingertip of their non-dominant hand by using a disposable Medlance® Red lancet-spike (HTL-Zone, Berlin, Germany) with a 1.5-mm blade and 2.0-mm penetration depth. A total of 300 µL of capillary blood was collected in a Microvette® CB 300 Z tube (Sarstedt, Nümbrect, Germany) with a clotting activator.

The blood samples were centrifuged (1,500$g$, 4 °C, 10 min) for serum separation. Creatine kinase (CK; EC 2.7.3.2) activity was determined in freshly separated serum with the use of the Accent 220S biochemical analyzer (Cormay, Łomianki, Poland) and sets of enzymatic reagents (Cat No. 7-220; Cormay, Łomianki, Poland). The rest of the collected serum (approximately 150 µL) was frozen at −80 °C until analysis.

Concentrations of serum cortisol (COR; Cat No. EIA-1887; DRG MedTek, Warsaw, Poland), serotonin (SER; Cat No. BA E-8900; LDN, Nordhorn, Germany), and brain-derived neurotrophic factor (BDNF; Cat No. 201-12-1303; Sunred Biological Technology, Shanghai, People's Republic of China) were determined using commercially available enzyme-linked immunosorbent assay (ELISA) kits. Spectrophotometric measurements of ELISA kits were performed on a multimode microplate reader (Synergy 2 SIAFRT; BioTek Instruments, Winooski, VT, USA).

The sensitivity of the methods for serum CK, COR, SER, and BDNF were 7.4 U/L, 2.5 ng/mL, 6.2 ng/mL, and 0.05 ng/mL respectively. The coefficients of variation (CVs) for the CK assay (0.92% and 3.28%), COR kit (3.2% and 7.7%), SER kit (12.6% and 10.4%), were respective to the intra- and inter-assay CVs. The intra-assay CV for BDNF was <10%.

## Training loads

Each player's RPE was separately collected roughly 20 min after each training session by using the CR-10 Borg scale as modified by *Scott et al. (2013)*. RPE was obtained by asking each player, "How hard was your session?" where 1 represented extremely easy and 10 represented maximum effort. All players were fully familiar with the scale range. The training load for each session (sRPE) was then calculated by multiplying the training duration (min) by the RPE as described by *Scott et al. (2013)*.

Training loads of the sessions were summed to provide data for 7 weeks of training. The cumulative data for a given training period (T0–T1, T1–T2, and T2–T3) included all training sessions and matches (friendly and championships) played in the respective training period.

## Characteristics of training loads during the pre-season and competitive periods

The 7-week winter pre-season period focused on the improvement the players' physical abilities. In addition to the winter training loads, the players also played four friendly matches. The principal elements of their training sessions during the winter pre-season period are shown in Table 1. Furthermore, a general schedule for the competitive period is presented in Table 2. In total, during the 14-week competitive period, the team played 13 championship matches.

## Statistical analysis

The sample size of each variable was 18 ($n = 18$). All data are presented as means ± standard deviations. All pairs of variables were examined for bivariate normal distribution using the Henze–Zirkler test (HZ) (*Henze & Zirkler, 1990*). A Pearson's correlation test was performed on normal variables.

Significance between means of variables was calculated using a Kruskal–Wallis test ($H^{KW}$) or Friedman test ($H^F$). Where significant differences appeared, Welch (for normal variables) and Wilcoxon tests were used. Cohen's (d) was calculated, and the effect sizes can be described in line with the following explanations: 0.35 or less = small effect size, between 0.35 and 0.65 = medium effect size, and 0.65 or larger = large effect size (*Cohen, 1988*). The level of statistical significance was set at alpha = 0.05 for all tests. All statistical analyzes were done using R programming language (version 4.0.2) (*R Core Team, 2020*).

## RESULTS

The Henze–Zirkler test showed that all pairs (T0, T1), (T0, T2), (T0, T3) of BDNF variable don't have bivariate normal distribution. For this biochemical marker (HZ = 0.945; $p = 0.01$), (HZ = 1.941; $p < 0.001$) and (HZ = 1.944; $p < 0.001$), were found, respectively. For other biochemical markers bivariate normal distributions were assumed. COR variable obtained (HZ = 0.516; $p = 0.21$), (HZ = 0.614; $p = 0.11$) and (HZ = 0.439; $p = 0.34$). For CK (HZ = 0.365; $p = 0.51$), (HZ = 0.491; $p = 0.24$) and (HZ = 0.713; $p = 0.05$) was found. SER variable obtained (HZ = 0.516; $p = 0.21$), (HZ = 0.163; $p = 0.98$) and (HZ = 0.164; $p = 0.98$), respectively.

Andrzejewski et al. (2022), *PeerJ*, DOI 10.7717/peerj.13367

**Table 1 General characteristics of training loads carried out during the pre-season period in the field of physical preparation.**

| Training characteristics | Week | Objective | Method | Form | Intensity | Volume (Duration) | Duration of effort | Repetitions | Set | Rest period | Sessions/week |
|---|---|---|---|---|---|---|---|---|---|---|---|
| Aerobic capacity/power | 1,2 | Aerobic capacity | Continuous | Moderate running | 70–85% MHR | 15–60 min | 10–15 min | 3–6 | 2–4 | Active Return to HR of 120 | 3–4 |
| | | | Long and medium interval | Technical/tactical exercise | | | | | | | |
| | | | Variable tempo | Game on large or medium pitch | | | | | | | |
| | | | | Technical circuit course | | | | | | | |
| | 3,4,5 | Aerobic power | Variable tempo | Intense running | 85–95% MHR | 10–40 min | 2–5 min | 2–6 | 2–4 | Active-semi-active 1:1 to 1:2 (W:R) | 2–3 |
| | | | Medium and short interval | Technical/tactical exercise | | | | | | | |
| | | | Intermittent | SSGs (3 × 3/4 × 4) | | | | | | | |
| Anaerobic capacity/power | 5,6 | Glycolytic capacity | Medium and short interval | Intense running Sprinting Technical work under pressure SSGs (1 × 1/2 × 2) | 95–100% MHR | 8–15 min | 30 s - 2 min | 3–5 | 1–2 | Passive-semi-active 1:1 to 1:3 (W:R) Between sets: 10 min | 1–2 |
| | 6,7 | Glycolytic power | Short interval with repetition | Sprinting Shuttle running | 95–100% MHR of max. speed | 600–1,000 m | 8–30 s | 3–4 | 2–3 | Active-semi-active 1:3 to 1:6 (W:R) Between sets: 8–10 min | 1 |
| Speed | 1,2 | Play speed | Play | Game on large or medium pitch | 80–90% max speed | – | – | – | – | – | 1 |
| | 3,4,5 | Speed - power | Short interval with repetitive | Sprinting Low jumps/high jumps Running games | 100% max speed | 200–400 m | less 6 s 10–40 m | 4–6 | 3–5 | Passive-semi-active 1:10 to 1:20 (1′ to 3′) Between sets: 4–6 min | 1–2 |
| | 6,7 | Speed - power | Short interval with repetitive | Sprinting High jumps | 100% max speed | 200–300 m | less 5 s 10–30 m | 4–5 | 2–4 | Passive-semi-active 1:10 to 1:15 (1′ to 2′) Between sets: 2–4 min | 1–2 |
| Strength | 1,2 | Strength endurance | Interval | Circuit-strength training or station work | 40–50% max strength | 30–45 min | 20–30 s | 15–20 | 3-4 | Passive-semi-active Between sets: 1–2 min | 2–3 |
| | 3,4 | Muscle hypertrophy | Interval | Station training (with loads) program with a personalised | 70–85% max strength | 30-40 min | – | 6–10 | 3–4 | Passive-semi-active Between sets: 2–3 min | 1–2 |
| | 4,5 | Strength maximum | Intermuscular coordination, Repetitive | Station training (with loads) program with a personalised | 80–100% max strength | 30–40 min | – | 1–5 | 4–6 | Passive-semi-active Between sets: 3–5 min | 2–3 |
| | 6,7 | Speed - strength | Repetitive | Station training (with loads) | 50–70% max strength | 30 min | – | 2–5 | 3–5 | Passive-semi-active | 2 |

**Note:**
MHR, maximum heart rate; SSGs, small-sided games; W:R, work to recovery time ratio.

**Table 2 Training program executed in a standard microcycle during the competitive period.**

Training microcycle in competitive period

| Training day | Training characteristics | Intensity T1–T2 | Duration of training | Intensity T2–T3 | Duration of training |
|---|---|---|---|---|---|
| Monday | Active recovery training + aerobic capacity | 70–85% MHR | 60–75 min | 60–70% MHR | 50–60 min |
| Tuesday | Power and technical and tactical training | 50–70% max strength, 85–95% MHR | 90 min | 50–60% max strength, 85–90% MHR | 80 min |
| Wednesday | Strength/Functional training + speed endurance | 95–100% MHR of max. speed | 90 min | 95–100% MHR of max. speed | 90 min |
| Thursday | Speed and power | 100% max speed | 70 min | 100% max speed | 60 min |
| Friday | Speed reaction and tactics training | 70–85% MHR | 70 min | 60–75% MHR | 70 min |
| Saturday | Game | 85–95% MHR | 90 min | 85–95% MHR | 90 min |
| Sunday | Free/passive recovery | – | – | – | – |

**Note:**
MHR, maximum heart rate; T1, end of the preparation period; T2, mid-competitive period; T3, end of the competitive period.

To check independence between bivariate normal variables Pearson's correlation test was used. To check linear independence for BDNF Kendall's correlation test was used. Only for CK all $p$-values greater than 0.05 were obtained ($t = 0.518$ (df = 16); $p = 0.61$), ($t = -1.230$ (df = 16); $p = 0.24$) and ($t = 0.858$ (df = 16); $p = 0.40$). For this reason Kruskal–Wallis test for this variable was used. For other Friedman test was performed, because at least one pair was significantly dependent, where other pairs have p-values slightly greater than 0.05. For COR ($t = 2.086$ (df = 16); $p = 0.05$), ($t = 2.413$ (df = 16); $p = 0.03$) and ($t = 1.596$ (df = 16); $p = 0.13$) was found. SER variable obtained ($t = 1.837$ (df = 16); $p = 0.09$), ($t = 2.674$ (df = 16); $p = 0.02$) and ($t = 2.632$ (df = 16); $p = 0.02$). For BDNF ($T = 124$; $p < 0.001$), ($T = 134$; $p < 0.001$) and ($T = 133$; $p < 0.001$) was found, respectively.

The statistical analysis of biochemical marker levels in relation to their various time measurements (T0, T1, T2, T3), revealed that an effect was observed for COR ($H^F = 12.800$ (df = 3); $p = 0.01$), CK ($H^{KW} = 15.237$ (df = 3); $p = 0.00$), SER ($H^F = 8.467$ (df = 3); $p = 0.04$) and BDNF ($H^F = 8.067$ (df = 3); $p = 0.04$). For COR, CK and SER paired Welch $t$-test was used. For BDNF paired Wilcoxon signed-rank test was used. Paired tests were performed, because repeated measures were done on the same participants. Results of abovementioned tests are illustrated in Table 3.

For sRPE and differences of biochemical marker levels Henze–Zirkler test was performed (in relation to their subsequent measurements (T0–T1, T1–T2, T2–T3)). Performed analyses showed that only pair [BDNF(T0–T1), BDNF(T1–T2)] does not have bivariate normal distribution (HZ = 1.627; $p < 0.001$). Other pairs [(T0–T1), (T2–T3)], [(T1–T2), (T2–T3)] had (HZ = 0.379; $p = 0.48$) and (HZ = 0.542; $p = 0.17$), respectively. For sRPE (HZ = 0.385; $p = 0.46$), (HZ = 0.609; $p = 0.11$) and (HZ = 0.349; $p = 0.56$) was found. COR variable obtained (HZ = 0.694; $p = 0.06$), (HZ = 0.596; $p = 0.12$) and (HZ = 0.296; $p = 0.71$). For CK (HZ = 0.362; $p = 0.52$), (HZ = 0.276; $p = 0.77$) and

**Table 3  The biochemical markers in studied youth soccer players at four measurement points and the relations to their baseline measurements (mean ± SD).**

| Parameters | Time points [mean ± standard deviation] | | | | p-values; effect size | | |
|---|---|---|---|---|---|---|---|
| | T0 | T1 | T2 | T3 | T0 *vs* T1 | T0 *vs* T2 | T0 *vs* T3 |
| COR (ng/mL) | 165.40 ± 39.40 | 221.62 ± 68.80 | 208.54 ± 81.97 | 218.97 ± 55.34 | $p = 0.001$ $d = 1.00$ | $p = 0.018$ $d = 0.67$ | $p = 0.001$ $d = 1.12$ |
| CK (U/L) | 151.39 ± 41.53 | 218.67 ± 42.51 | 201.33 ± 53.96 | 193.56 ± 66.03 | $p = 0.000$ $d = 1.60$ | $p = 0.004$ $d = 1.04$ | $p = 0.029$ $d = 0.76$ |
| SER (ng/mL) | 146.85 ± 66.66 | 187.55 ± 72.42 | 163.97 ± 72.21 | 144.57 ± 68.04 | $p = 0.035$ $d = 0.58$ | ns | ns |
| BDNF (ng/mL) | 9.85 ± 5.45 | 11.86 ± 6.43 | 9.79 ± 6.33 | 9.90 ± 6.22 | $p = 0.043$ $d = 0.34$ | ns | ns |

**Note:**

COR, serum cortisol concentration; CK, serum creatine kinase activity; SER, serum serotonin concentration; BDNF, serum brain-derived neurotrophic factor concentration; T0, start of the preparation period; T1, end of the preparation period; T2, mid-competitive period; T3, end of the competitive period; ns, not significant.

(HZ = 0.234; $p = 0.87$) was found. SER variable obtained (HZ = 0.243; $p = 0.85$), (HZ = 0.157; $p = 0.98$) and (HZ = 0.381; $p = 0.47$), respectively.

For all bivariate normal variables Pearson's correlation test was performed. For all pairs [(T0–T1), (T2–T3)] $p$-values much greater than 0.05 were obtained: sRPE ($t = -0.363$ (df = 16); $p = 0.72$), COR ($t = -0.492$ (df = 16); $p = 0.63$), CK ($t = -0.033$ (df = 16); $p = 0.97$), SER ($t = 0.080$ (df = 16); $p = 0.94$) and BDNF ($t = 0.046$ (df = 16); $p = 0.96$). From above results Kruskal–Wallis test on all variables was carried out. For other pairs [(T0–T1), (T1–T2)] and [(T1–T2), (T2–T3)]: sRPE – ($t = 0.733$ (df = 16); $p = 0.47$) and ($t = 1.491$ (df = 16); $p = 0.16$), COR – ($t = -1.636$ (df = 16); $p = 0.12$) and ($t = -4.395$ (df = 16); $p < 0.001$), CK – ($t = -1.626$ (df = 16); $p = 0.12$) and ($t = -2.556$ (df = 16); $p = 0.02$), SER – ($t = -2.612$ (df = 16); $p = 0.02$) and ($t = -2.313$ (df = 16); $p = 0.03$), were found, respectively. In case of BDNF Pearson's correlation test only on pair [(T1–T2), (T2–T3)] was performed and ($t = -1.296$ (df = 16); $p = 0.21$) was found.

The statistical analysis of sRPE and differences of biochemical marker levels revealed that a significant effect was observed for all the variables: sRPE ($H^{KW} = 13.189$ (df = 2); $p = 0.00$), COR ($H^{KW} = 9.261$ (df = 2); $p = 0.01$), CK ($H^{KW} = 12.492$ (df = 2); $p = 0.00$), SER ($H^{KW} = 7.781$ (df = 2); $p = 0.02$) and BDNF ($H^{KW} = 15.160$ (df = 2); $p < 0.001$) as illustrated in Fig. 2 and Table 4.

## DISCUSSION

The main objective of this study was to assess the impact of the applied training loads on the concentration of indicators characterizing the level of training-related stress in youth soccer players. Moreover, an attempt was made to determine the relation between the sRPE and the abovementioned biochemical parameters. The main findings were that a positive relation between the applied training loads measured with sRPE and the markers of training stress in youth soccer players were found.

1. The most demanding training loads applied in the preparation period (highest sRPE values) resulted in a significant increase in all analyzed biochemical training stress indicators.

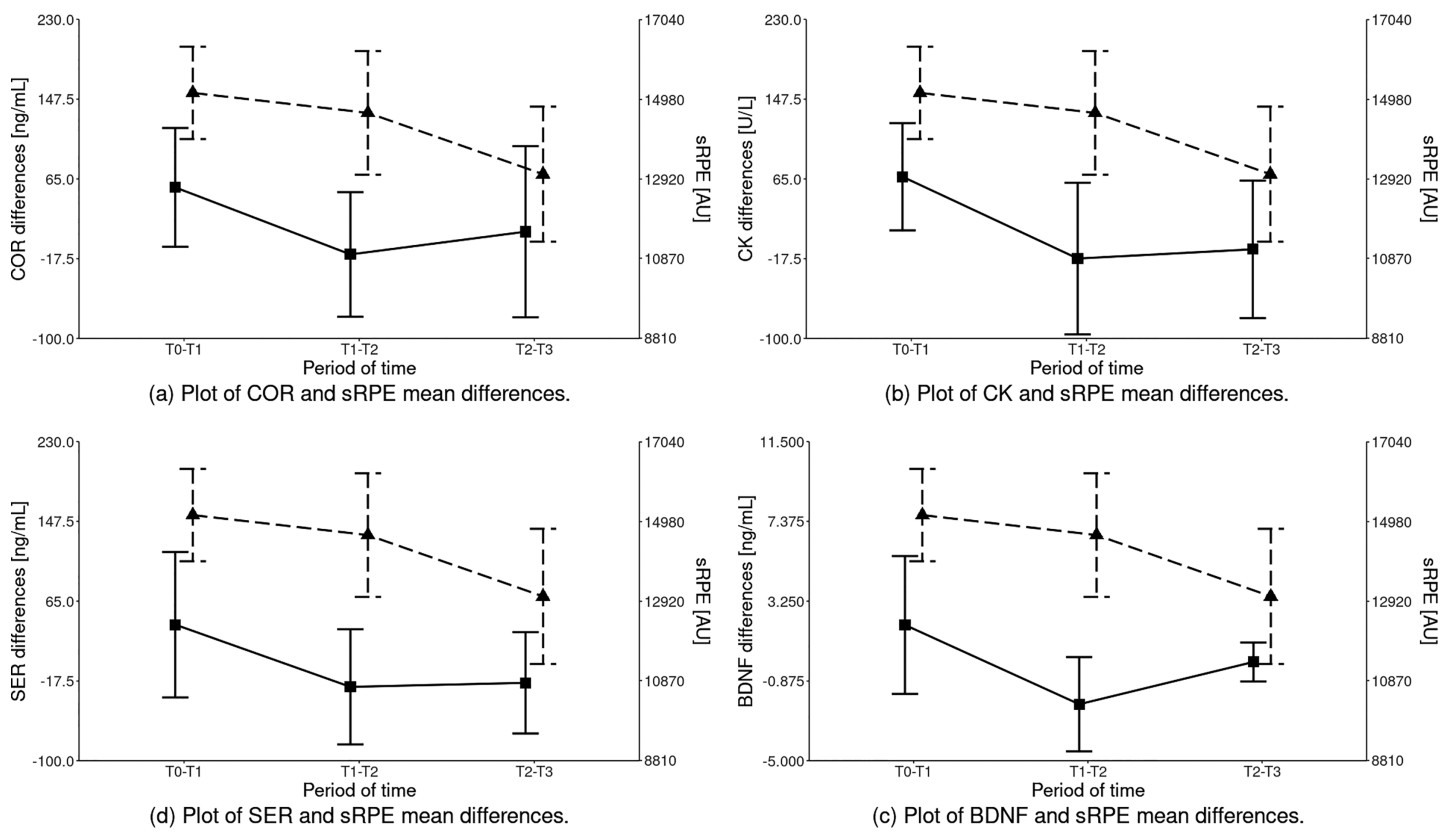

**Figure 2 Inter-term differences of the analyzed parameters depending on the training period in relation to sRPE (mean ± SD). Differences statistically significant:** sRPE – T1–T2 *vs* T2–T3, T0–T1 *vs* T2–T3; COR – T0–T1 *vs* T1–T2; CK – T0–T1 *vs* T1–T2, T0–T1 *vs* T2–T3; SER – T0–T1 *vs* T1–T2, T0–T1 *vs* T2–T3; BDNF – T0–T1 *vs* T1–T2, T1–T2 *vs* T2–T3, T0–T1 *vs* T2–T3. COR, serum cortisol concentration; CK, serum creatine kinase activity; SER, serum serotonin concentration; BDNF, serum brain-derived neurotrophic factor concentration.

**Table 4 Inter-term differences of the analyzed parameters depending on the training period (mean ± SD).**

| Parameters | Training period (mean ± standard deviation) | | | *p*-values; effect size | | |
|---|---|---|---|---|---|---|
| | T0–T1 | T1–T2 | T2–T3 | T0–T1 *vs* T1–T2 | T1–T2 *vs* T2–T3 | T0–T1 *vs* T2–T3 |
| sRPE (AU) | 15,148.72 ± 1,192.61 | 14,633.00 ± 1,597.87 | 13,048.06 ± 1,747.08 | ns | $p = 0.008$ $d = 0.95$ | $p = 0.000$ $d = 1.40$ |
| COR (ng/mL) | 56.22 ± 61.48 | −13.08 ± 64.46 | 10.42 ± 88.45 | $p = 0.002$ $d = 1.10$ | ns | ns |
| CK (U/L) | 67.28 ± 55.48 | −17.33 ± 78.45 | −7.78 ± 71.08 | $p = 0.001$ $d = 1.25$ | ns | $p = 0.001$ $d = 1.18$ |
| SER (ng/mL) | 40.70 ± 75.22 | −23.58 ± 59.56 | −19.40 ± 52.41 | $p = 0.008$ $d = 0.95$ | ns | $p = 0.009$ $d = 0.93$ |
| BDNF (ng/mL) | 2.02 ± 3.56 | −2.08 ± 2.44 | 0.11 ± 1.01 | $p = 0.000$ $d = 1.34$ | $p = 0.002$ $d = 1.17$ | $p = 0.041$ $d = 0.73$ |

**Note:**
COR, serum cortisol concentration; CK, serum creatine kinase activity; SER, serum serotonin concentration; BDNF, serum brain-derived neurotrophic factor concentration; T0, start of the preparation period; T1, end of the preparation period; T2, mid-competitive period; T3, end of the competitive period; ns, not significant.

2. The reduction in the training loads during a competitive period and the addition of recovery training sessions resulted in a systematic decrease in the values of the measured biochemical indicators.

During times of increased training load and less than optimal recovery, elevated resting cortisol can be detected due to disruption in homeostasis and the subsequent stress response (*Michailidis, 2014*; *Meeusen et al., 2013*; *Anderson & Wideman, 2017*). This was also the case in our study. The implementation of training loads during the pre-season and competitive period contributed to a significant increase in the players' blood cortisol levels. It is suggested that the performance of consecutive high-demanding exercises could be related to rising cortisol levels (*Tremblay, Copeland & Van Helder, 2004*; *Ratamess et al., 2005*). This phenomenon was reflected in our study where, during the pre-season period (T0–T1), the highest sRPE values and COR concentrations were recorded. Studies analyzing the relationship between the long-term training load and assessment of cortisol concentrations should receive more attention (*Skoluda et al., 2012*). The present study reveals a positive correlation between the cortisol concentration and training volume, which is characteristic for a soccer pre-season period.

It is worth emphasizing that in our study, the concentration of cortisol remained unchanged from the end of the pre-season period (T1) to the end of the competitive period (T3) (Table 3). This finding agrees with results of *Moreira et al. (2016)* who investigated the subsequent impact of a congested period of match play on resting hormonal parameters in elite youth soccer players. Based on these collective results, and by considering the resting COR values, it can be speculated that the competitive period might not significantly affect the hypothalamic-pituitary-adrenal axis response in youth soccer players.

On the other hand, a study by *Michailidis (2014)* reported that the highest cortisol concentrations were detected during the competitive season and that this may have been due to an accumulation in high stress levels during the league games. Results from our study contrasted with this finding, as COR levels were maintained during the competitive period when youth players had reduced training loads but participated in matches of the championship. They also presented with lower values of sRPE in T1–T2 and T2–T3 compared to T0–T1. The contrast in results might be explained by the fact that mental stress levels in youth soccer players are most likely at a lower level than that of professional players during a competitive period.

Another phenomenon frequently associated with both soccer training and match loads, is that of skeletal muscle microtrauma which may contribute to the development of inflammation and persistent fatigue in players (*Romagnoli et al., 2016*; *Souglis et al., 2015*). In our study, a significant increase in the resting CK levels was observed for all measurement points in relation to the baseline measurement (T0). Similarly, *Becatti et al. (2017)* found a significant increase in CK activity at the start of the competitive period relative to the pre-season period. They indicated that CK activity returned to basal levels in the middle of the competitive period before again sharply increasing toward the end of the

competitive period. They also associated fluctuations of CK levels with sRPE values for training loads.

*Silva et al. (2014)* reported significant increases in CK values in the middle (more than two-fold) and at the end of the competitive season when compared to CK values at the beginning of the pre-season period. They also reported a sharp decrease in off-season CK levels (back to baseline levels when training loads are reduced) compared to CK levels during the competitive period. The same authors stated that decreasing weekly training loads during competitive periods could contribute to decreasing CK levels and COR concentrations by the end of this training period. In our study, we did not observe any changes in CK levels during the competitive period (T2 *vs* T3). It is reasonable to presume that a reduction in the volume of training sessions during a competitive period and the addition of recovery training sessions might aid in decreasing muscle demands and thereby preventing the development of muscle damage in youth soccer players. Recovery training sessions might especially be useful in the second part of the competitive period (T2–T3), when we recorded the lowest sRPE values. Notably, *Aquino et al. (2016)* suggested that different training strategies, such as a training periodization with emphasis on technical-tactical abilities, could be implemented to prevent or help correct situations of accumulated muscle microtrauma in youth soccer players that form part of a competitive context.

Up until the early 2000s, sports scientists mainly focused on peripheral, "muscle" training-related stress factors. More presently, explanations of training stress have also been based on the involvement of the central nervous system (CNS) in this process. Disturbances in neurotransmitter syntheses in CNS can manifest as psychological loss of motivation, loss of appetite, sleepiness, reduced concentration or cognitive function (*Morris et al., 2015*).

Many researchers have proven that accumulation of serotonin in different brain regions is contributed to the development of fatigue during prolonged exercise (*Newsholme, Acworth & Blomstrand, 1987*; *Meeusen, Watson & Dvorak, 2006*). Elevated serum serotonin concentrations were recorded after high intensity or prolonged exercise efforts, such as an ultramarathon (*Agawa et al., 2008*), 35 min of running with high intensity (85% HRmax) (*Zimmer et al., 2016*), as well as exhaustive exercise in the heat (*Zhao et al., 2015*). One study reported that the use of specific activating serotonergic pathways dramatically decreased physical performance and increased rating of perceived exertion during and after the exercise (*Marvin et al., 1997*). Contrastingly, many researchers have shown that decreased serotonergic activity affects higher exercise performance and lower perceived exertion (*Knechtle et al., 2012*; *Teixeira-Coelho et al., 2014*; *AbuMoh'd, Matalqah & Al-Abdulla, 2020*).

There are, however, very few studies on the impact that training might have on serotonin levels (adding value to our research outputs in this regard). To our knowledge, our research provides the only report that covered such a long period, and wherein the concentration of serotonin as a marker of training-related stress was measured up to four times. Our analyses highlighted a significant increase in the mean values of this indicator, but only after the most exhaustive training sessions which took place in the

pre-season period (and showed the highest sRPE in the T0–T1 period). With subsequent measurement points in the competitive period (T2 and T3), we recorded a systematic, although statistically insignificant decrease in the concentration of this indicator. This may be explained by the reduction of training loads during the competitive period (Table 2). Corresponding to the decrease in sRPE in subsequent training periods, a reduction in serotonin increases was also noted. In comparison to T0–T1, serotonin values in T1–T2 and T2–T3 periods were lower. This may be explained by the fact that only excessive endurance training, which was used in the pre-season period, was a strong enough stimulus to significantly affect the blood serotonin concentration.

As already mentioned, training stress is a complex process. *Davis & Bailey (1997)* found that increased serotonin concentration during physical exercises inhibits dopamine, which is involved in movement initiation (*Chaudhuri & Behan, 2000*) and therefore results in reduced physical performance. The external regulator of dopamine is not only serotonin but also BDNF. The role of BDNF is of course not limited to the regulation of the dopaminergic system. This biochemical factor is also involved in many physiological processes such as glycogen homeostasis and lipid metabolism (*Jiménez-Maldonado et al., 2018*). Increased circulating BDNF concentrations have been reported following an acute bout of both aerobic and resistance exercise (*Jiménez-Maldonado et al., 2018*; *Slusher et al., 2018*) as well as the recovery period, thus appearing to be associated with both duration and intensity of exercise (*Saucedo Marquez et al., 2015*; *Church et al., 2016*). *Seifert et al. (2010)* reported that endurance training could enhance BDNF release from the human brain whilst, *Jeon & Ha (2015)* indicated that long-term regular aerobic training also increased the serum BDNF level at rest. In addition, similar changes to those after aerobic training were noted after 3-months of whole-body CrossFit training (*Murawska-Cialowicz, Wojna & Zuwala-Jagiello, 2015*).

In contrast, decreased BDNF concentrations have been reported for soldiers that follow very intense military training (*Gepner et al., 2019*), whilst *Figueiredo et al. (2019)* noted lower BDNF levels following 8-weeks of high-intensity endurance training combined with strength training. *Murawska-Ciałowicz et al. (2021)* recently reported that resting values of BDNF remained unchanged or were reduced after different forms of high-intensity training for 9-weeks and *Hebisz et al. (2019)* found no changes in BDNF after 6-months of long-term sprint interval training in well-trained cyclists.

The findings from our research were similar to that of *Seifert et al. (2010)* and *Jeon & Ha (2015)*, in that we also detected an increase in resting values of BDNF concentrations. Statistically significant increases in these values, compared to that of the baseline, were only observed after the pre-season training period, when the training was the most excessive. After the pre-season period, we also noted a significant increase in resting values of COR, CK, and SER. Interestingly, while our research contradicts the BDNF concentration findings of *Gepner et al. (2019)*, *Figueiredo et al. (2019)* and *Murawska-Ciałowicz et al. (2021)* the lower sRPE values measured in the subsequent training periods of our study were in relation to lower differences in BDNF concentration.

Interestingly, the increase in resting values of BDNF in the pre-season period (T0–T1) of our study was in relation to higher CK activity (muscle damage indicator), COR
(physical and psychological stress indicator), and SER (central training-related stress indicator) concentrations. Some research (*Clow & Jasmin, 2010*; *Gu, Ding & Williams, 2014*) showed that possible physiological mechanism associated with the decreased BDNF concentration is based on the usage of this substrate in the regeneration of myo- and nerve fibers in damaged tissues. Additionally, metabolic stress can also decrease BDNF secretion.

One study (*de Assis & Gasanov, 2019*) reported a negative correlation between BDNF and the cortisol level, whereas *Garcia-Suarez et al. (2020)* found no change in BDNF accompanied by a higher COR level and COR/BDNF ratio. *Verbickas et al. (2017)* noted that after sprint interval training, serum CK activity as well as BDNF and COR levels increased significantly (and after 24-h again decreased significantly). They also noted a positive correlation between BDNF levels and central activation ratio, suggesting that lower BDNF concentration is associated with a lower level of stress related to exercise. Our own findings can also suggest that BDNF, similar to markers such as serotonin, can be used as a central training-related stress indicator.

### Limitations and future studies

As an observational study, the size of the study group was limited and therefore findings for a larger population will have to be assessed. For the same reason, the studied variables might follow a different trend in a population with subjects originating from different sports and regions. It is recommended to conduct further research using other parameters characterizing the internal training loads (*e.g.* HR monitoring, acute: chronic workload ratio - ACWR) and data from GPS as parameters characteristic for the external training loads. Correlating data such as GPS training and match running performance with biochemical training-related stress markers and sRPE may shed new and valuable light on the control of training loads and the performance of soccer players during the training program.

## CONCLUSIONS

In conclusion our research showed a positive relation between the applied training loads measured with sRPE and the markers of training stress in youth soccer players. Only the most demanding training loads applied in the preparation period (highest sRPE values) resulted in a significant increase in all analyzed biochemical training stress indicators compared to baseline. Reduction in the training volume and the addition of recovery training sessions during a competitive period, when high-intensity training dominates, resulted in a systematic decrease in the values of the measured biochemical indicators. Our research showed also that a combination of subjective and objective markers (including training loads) is useful in monitoring aspects of training stress in youth soccer players. The combination of these markers will provide coaches with more sufficient evidence to appropriately tailor training and recovery periods for an individual player and thereby allowing for their optimized performance.

## ACKNOWLEDGEMENTS

The authors would like to thank the athletes who took part in this study.

### Funding

This research was funded by a grant from the National Science Centre - no. DEC-2017/01/X/NZ7/00336 MINIATURA, which is a scheme offering funding for activities that serve as part of larger basic research. The funders had no role in study design, data collection and analysis, decision to publish, or preparation of the manuscript.

### Grant Disclosures

The following grant information was disclosed by the authors:
National Science Centre: DEC-2017/01/X/NZ7/00336 MINIATURA.

### Competing Interests

The authors declare that they have no competing interests.

### Author Contributions

- Marcin Andrzejewski conceived and designed the experiments, performed the experiments, analyzed the data, prepared figures and/or tables, authored or reviewed drafts of the paper, and approved the final draft.
- Marek Konefał analyzed the data, prepared figures and/or tables, authored or reviewed drafts of the paper, and approved the final draft.
- Tomasz Podgórski conceived and designed the experiments, performed the experiments, analyzed the data, prepared figures and/or tables, authored or reviewed drafts of the paper, and approved the final draft.
- Beata Pluta performed the experiments, prepared figures and/or tables, and approved the final draft.
- Paweł Chmura analyzed the data, authored or reviewed drafts of the paper, and approved the final draft.
- Jan Chmura conceived and designed the experiments, authored or reviewed drafts of the paper, and approved the final draft.
- Jakub Marynowicz performed the experiments, prepared figures and/or tables, and approved the final draft.
- Kamil Melka analyzed the data, prepared figures and/or tables, and approved the final draft.
- Marius Brazaitis analyzed the data, authored or reviewed drafts of the paper, and approved the final draft.
- Jakub Kryściak conceived and designed the experiments, performed the experiments, analyzed the data, prepared figures and/or tables, authored or reviewed drafts of the paper, and approved the final draft.

## Human Ethics

The following information was supplied relating to ethical approvals (*i.e.*, approving body and any reference numbers):

Poznan University of Medical Sciences approval to carry out the study within its facilities (no. 973/17).

## Data Availability

The raw measurements are available in the Supplemental File.

## Supplemental Information

Supplemental information for this article can be found online at http://dx.doi.org/10.7717/peerj.13367#supplemental-information.

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
