# Peer review of "How training loads in the preparation and competitive period affect the biochemical indicators of training stress in youth soccer players?"

_PeerJ, doi:10.7717/peerj.13367_

## Round 0.1 · original submission · Major Revisions

Dear Authors,

Please make corrections to your manuscript or write a detailed rebuttal on a point-by-point basis regarding the reviewers' comments.

Reviewer 1 ·

Basic reporting

Overall I thought the manuscript/study design was interesting. I believe this data will add some unique data that will spark a lot of conversations and future studies. I so have some minor suggested edits. see below:

METHODS
Line 161-162 why was a 36 hour time period selected?
Lines 215-219 why was creatine kinase activity determined separately?
Line 252 Should “size of every variable…” be rephrased? The sample size of each variable was 18(n = 18)?
RESULTS
Lines 265-267 I think this was lost in translation
Lines 269-270 Same thing. Change “was get” to “was found”
Lines 272-273 Is this information that belongs in the methods?
Line 278 Change “was get” to “was found”
Line 280 Change “was get respectively” to “was found, respectively”
Line 291 Performed analyses? Change to “performed analyses”
Line 292 Change “doesn’t” to “does not”
Line 295 Change “was get” to “was found”
Line 306 Change “was get” to “was found”
Line 308 Change “was get” to “was found”
DISCUSSION
Mainly grammatical concerns
Line 410 change to “8 weeks of resistance”
Line 417 Change “analyzes” to “analyses”

Experimental design

no comment

Validity of the findings

no comment

Additional comments

no comment

Reviewer 2 ·

Basic reporting

The structure, figure and tables are sufficient.

Experimental design

Experimental design is well done

Validity of the findings

This research is valid.

Additional comments

General Comments

This is an interesting study, assessing the changes in blood biochemical markers and rate of perceived exertion (RPE) in young male soccer players (n=18, mean age 18 years) following 6-month training period (January-June). Blood biochemical markers (cortisol, creatine kinase, serotonin, and brain-derived neurotrophic factor – BDNF) were measured four times: at the beginning of the preparation period, immediately after the preparation period, in the middle of the competitive period, and at the end of the competitive period. Blood samples were taken from players who had been out of training for about 36 hours (or 36 hours after training). The training loads were assessed using a session RPE, and the assessment time points were chosen so that fatigue after a soccer match would not be assessed.

The main findings of this study were that: (1) after the pre-season (most demanding training) period the values of all measured blood biochemical markers increased significantly; (2) the reduction of training loads in the competitive period resulted in a systematic decrease in the values of the measured blood biochemical markers; (3) the one exception was the BDNF concentration, which increased significantly. The latter could be related to mental and physical stress accrued over the course of the training time.
This text is well-written in general. However, before it is published, the design of this work should be improved.

1. I suggest not to use the term "indicators central fatigue (assessed by serotonin and BDNF) and peripheral fatigue (assessed by cortisol and creatine kinase)" in context of this study in Title and for explanation of the results of this study in Discussion and Conclusion. Fatigue during prolonged exercise has been defined as the inability to maintain the required or expected muscle force/power output that leads to a loss of performance in a given task (Edwards RH. 1981). This definition is generally accepted by exercise physiologist. In this study, the changes in force-generation capacity of skeletal muscles or in physical of performance (fatigue and recovery) were not studied. Neuromuscular (physical) fatigue is indicator of acute (short-time) adaptation to exercise. In this study, the long-term (chronic) adaptation to soccer training loads during 6-months period was assessed by 4 blood biochemical markers: cortisol – typical marker of chronic stress, creatine kinase – typical marker of muscle damage, serotonin – which is also marker of central fatigue, but only in complex with other central factors that might limit exercise performance (Meeusen R et al., Sports Med, 2006), and BDNF – participates in physiological processes such as glycogen homeostasis and lipid metabolism (Jimenez-Maldonado A et al. Front Neurosci, 2018). I suggest to use for explanation of the changes in blood biochemical markers in context of this study using terms "exercise-related stress", or "training stress". These terms more undestandable in context of long-term adaptation to training loads in young soccer playes.

2. Authors should better to explain the novelty of this study in Introduction because this is not the study of central and peripheral fatigue but study of physiological/biochemical adaptation to regular training loads in young soccer players.


Specific Comments

TITLE: Please correct the Title according to General Comments

ABSTRACT: Please correct text avoiding term "central and peripheral fatigue" (see General Comments)

INTRODUCTION: Please explain better the novelty of this study in context of long-term physiological/biochemical adaptation to regular training loads in young soccer players (see General Comments)

DISCUSSION: Please correct text in context of long-term physiological/biochemical adaptation to regular training loads in young soccer players (see General Comments)

CONCLUSIONS: please correct text according to General Comments

TABLE 2: Please add Abbreviations T1, T2 and T3 behind this table.

TABLE 3: Please add Abbreviations T0, T1, T2 and T3 behind this table.

TABLE 4: Please add Abbreviations T0, T1, T2 and T3 behind this table.

·

Basic reporting

The reviewer’s comments on the paper are as follows:

I have reviewed the research article entitled “How youth soccer training affects the indicators of peripheral and central fatigue?”
It is interesting topic and totally, this manuscript is well organized and written. However, there are just a few points of clarification that need to be addresses. Therefore, I recommend that the manuscript must be revised before it can be reconsidered for publication. Here are my comments:

- Page 6: The title doesn’t show in which training periods or mesocycles the fatigue markers were considered for investigation. So, please add these periods to the title (for example: preparatory and competitive period).
- Page 7, line 74-78: Discussion at the end of the abstract doesn’t fully reflect the title mean. It is advised to the authors to rewrite this with priority of training effectiveness and then to mention the relations between the respective variables.
- Page 7: Please add the word that represent an intervention including training in this study.
- Page 7, Introduction: Considering the youth sample was selected in this research, please explain the reasons for choosing them rather than adults.
- Page 8, Introduction: Serotonin and BDNF are acceptable as the central fatigue. However, it was better to measure or to add dopamine and calculating ratio of serotonin to dopamine as a more suitable indicator of CNS fatigue. Please clarify.
- Page 9, Materials and methods, line 187-189: the authors wrote “, their nutrition and hydration habits were well controlled, following the recommendations by the soccer center nutritional staff. The players followed the same nutritional and hydration protocol during the soccer season”. I didn’t see any information about it. Please report the subjects’ nutritional information composed of macronutrients distribution and total calorie intake as an individual table. These could have remarkable influences on the dependent variables measured in this study such as sensitive marker of cortisol.
- Page 25 & 27: Quality of figure 1 and 2 included in the PDF file is not appropriate and their size seems small. Please redraw them with more resolution and larger size.
- Page 13, Discussion: the results are repeated at the first lines of this section (line 325-340) and specified in the numbers. It is advised to the authors to remove them and discuss each set of findings individually (To mention first result and continuing with required explanations).
- Page 13-16, Discussion: All the p values throughout this section need to be removed. The authors reported them previously in the results.
- Page 17, Limitation: Please add the lack of control group at this study as another limitation.
- Page 17, Conclusions (line 481-486): the first 6 line are repetitive and redundant. Please rewrite briefly with emphasis on the principal points derived from the study.
- Page 17-23, References: The number of references is very high (85 ref) for a research paper. Many of them are relatively old and it is suggested to the authors to remove less relevant references as possible. Also, the authors could include below reference related to this study in the references and apply its content:
- Arazi H, Asadi A, Khalkhali F, Boullosa D, Hackney AC, Granacher U and Zouhal H (2020) Association Between the Acute to Chronic Workload Ratio and Injury Occurrence in Young Male Team Soccer Players: A Preliminary Study. Front. Physiol. 11:608. doi: 10.3389/fphys.2020.00608

Experimental design

Please refer to the basic reporting.

Validity of the findings

Please refer to the basic reporting.

Additional comments

Please refer to the basic reporting.

Reviewer 4 ·

Basic reporting

Please check it in pdf file

Experimental design

Please check it in pdf file

Validity of the findings

Please check it in pdf file

Additional comments

Please check it in pdf file

Annotated reviews are not available for download in order to protect the identity of reviewers who chose to remain anonymous.

---

## Round 0.2 · accepted · Accept

Dear authors,

After the corrections you made your manuscript is acceptable for publication in its current form.

Reviewer 2 ·

Basic reporting

No comment

Experimental design

No comment

Validity of the findings

No comment

Additional comments

The design of this manuscript was improved in process of review.